# A NIR-Fluorochrome for Live Cell Dual Emission and Lifetime Tracking from the First Plasma Membrane Interaction to Subcellular and Extracellular Locales

**DOI:** 10.3390/molecules29112474

**Published:** 2024-05-24

**Authors:** Eden Booth, Massimiliano Garre, Dan Wu, Harrison C. Daly, Donal F. O’Shea

**Affiliations:** Department of Chemistry, Royal College of Surgeons in Ireland (RCSI), D02 PN40 Dublin, Ireland

**Keywords:** dual near-infrared emission, BF_2_-azadipyrromethene, plasma membrane, intracellular vesicles, extracellular vesicles, fluorescence lifetime

## Abstract

Molecular probes with the ability to differentiate between subcellular variations in acidity levels remain important for the investigation of dynamic cellular processes and functions. In this context, a series of cyclic peptide and PEG bio-conjugated dual near-infrared emissive BF_2_-azadipyrromethene fluorophores with maxima emissions at 720 nm (at pH > 6) and 790 nm (at pH < 5) have been developed and their aqueous solution photophysical properties determined. Their inter-converting emissions and fluorescence lifetime characteristics were exploited to track their spatial and temporal progression from first contact with the plasma membrane to subcellular locales to their release within extracellular vesicles. A pH-dependent reversible phenolate/phenol interconversion on the fluorophore controlled the dynamic changes in dual emission responses and corresponding lifetime changes. Live-cell confocal microscopy experiments in the metastatic breast cancer cell line MDA-MB-231 confirmed the usability of the dual emissive properties for imaging over prolonged periods. All three derivatives performed as probes capable of real-time continuous imaging of fundamental cellular processes such as plasma membrane interaction, tracking endocytosis, lysosomal/large acidic vesicle accumulation, and efflux within extracellular vesicles without perturbing cellular function. Furthermore, fluorescence lifetime imaging microscopy provided valuable insights regarding fluorophore progression through intracellular microenvironments over time. Overall, the unique photophysical properties of these fluorophores show excellent potential for their use as information-rich probes.

## 1. Introduction

Fluorescence microscopy is a cornerstone technique for the investigation of cellular function and processes [1,2,3,4]. Most commonly, scientific data are obtained in the form of an image derived from the locations in which the fluorophore accumulates, though achieving dynamic image selectivity can be challenging from an always-on fluorophore. The possibility of time course imaging of this response becomes viable when a fluorophore is engineered to modulate fluorescence intensity, for example, from off-to-on [5,6,7]. Yet a distinct limitation of off-to-on fluorophores is that data are only obtainable in the on-state as, by default, the fluorescent silent off-state yields no information. A more data-rich construct than off-to-on responsive would be the counterintuitive sounding design of on-to-on, in which a fluorophore can exist in two interconvertible, response-differing, and discernable on-states [8,9,10,11,12,13,14,15,16]. In this way, data can be collected simultaneously from both on-states, doubling the number of data sources. Practically, two interconvertible on-states could be distinguished from each other by having differing emission wavelengths. The emissions may also have distinguishable lifetimes, allowing for fluorescence lifetime imaging microscopy (FLIM) to be employed, thereby providing an additional layer of data.

Introduced in the early 2000s, the BF_2_-azadipyrromethenes are an exciting near-infrared (NIR) emissive platform from which application-specific imaging agents can be constructed [17,18,19]. They are ideally suited to both live cellular and in vivo imaging, with fluorescence wavelengths tunable within the low-level energy range of 650 to 850 nm, low toxicity, and excellent photo stability [20,21,22,23,24,25,26,27]. Previously, we reported a BF_2_-azadipyrromethene of general structure A as a lysosomal acidity off-to-on responsive probe with excellent photostabilities, good quantum yields, and no phototoxicity at concentrations used for imaging (Figure 1) [28]. Its modulating construction differed from typical basic nitrogen-containing lysosomotropic stains, with its off-to-on fluorescence switch controlled by the conversion of a phenolate to phenol in the pH range of 6 to 4 [27,28]. As the *o*-nitro phenol, the fluorophore has an emission maximum of 720 nm, which is quenched upon deprotonation. This responsive single emission was employed for both live cell imaging and in vivo tumor imaging [29]. As the off/on switching mechanism is reversible, it is capable of real-time continuous imaging of acidic regions over prolonged periods without perturbing cellular function (Figure 1, previous work).

Fluorophores with the potential to elicit two distinct interchangeable and relatable emissions are of elevated value with respect to single signal emissions. This data-rich approach is especially useful in dynamically changing live cell imaging scenarios. In this work, we demonstrate how an individual NIR fluorochrome can be exploited for simultaneous intensity and lifetime imaging through two interconvertible wavelengths operational in the ranges of 715 (±25) and 810 (±40) nm and within a lifetime range from 0.7 to 1.2 ns (Figure 1, this work). From a spectroscopic understanding of the interconverting relationship of the two emissions, it has been possible in live cell imaging to identify first contact with the plasma membrane, inner leaf trafficking vesicles, trafficking through the intracellular milieu via endosomes and lysosomes, large acidic vesicles (LAV), and extracellular vesicles (EVs) produced by the cells.

## 2. Results and Discussion

It is known that deprotonation of simple phenolic compounds to the corresponding phenolate results in a bathochromic shift of absorbance and emission spectra. However, this property has not been widely explored for live-cell imaging [30,31,32]. For this work, it was anticipated that the as-of-yet unexplored phenolate state of type A fluorophores could have a distinct emission bathochromically shifted from the phenol form (Figure 1, this work). As both emissions would interrelate to each other in an on-to-on manner, one of the two interconvertible emissions would always be on, and, under specific circumstances, both may be on. If utilized in this format, the data loss imposed by a dark-off state could be circumvented. As such, we were encouraged to explore the prospect of expanding their use from one to two wavelengths using microscopy instrumentation capable of routine imaging in the 700–850 nm range [33,34].

The level of acidity within cells is strictly regulated to provide optimal activity of cellular processes and differs between various compartments [35,36,37]. The pathway(s) by which a molecular fluorophore is internalized and transported through the intracellular milieu can expose it to a sequence of relative acidities from pH 7.2 to 4.2. Using a dual emissive fluorophore, these differences could be spatially and temporally identifiable through dynamic recording of emission wavelengths and lifetimes [38,39,40]. This could include first contact of the fluorophore from the media with the cell plasma membrane, accumulation within early endosomes, lysosomes, and LAVs, and ultimately within EVs following their release from the cells. To investigate this possibility, three conjugates of structure A were used: two cyclic peptides containing RGD (**1**) or RAD (**2**) tripeptide sequences and one pegylated example (**3**) with ~111 repeating PEG units (Figure 2).

### 2.1. Fluorophores and Photophysical Characterizations

The synthesis of **1** and **3** have been previously reported, and **2** was produced using the same synthetic route [28]. At the outset, it was necessary to establish if an emission from the phenolate state of **1**–**3** was present and how it related to the corresponding known phenol emission. To mimic their responses in the aqueous extracellular medium and varying pH microenvironments found within live cells, the spectroscopic characteristics of **1**, **2**, and **3** were investigated using Dulbecco’s modified Eagle’s medium (DMEM) as a solvent with pH levels adjusted using 0.1M HCl to obtain solutions of pH 1, 4, and 7. As shown in Figure 3, it was observed that at relatively acidic environments of pH 1 and pH 4, the emission maxima of **1**–**3** overlapped and were at 720, 719, and 719 nm, respectively, corresponding to the phenol species (panels A–C red and black traces). In contrast, at pH 7, the emission maxima red-shifted to 787, 783, and 792 nm for **1**–**3**, respectively, corresponding to their phenolate species (panels A–C, green traces, Table 1). When the fluorophores were subjected to repeat cycles of acidification and neutralization, no loss in fluorescence intensity was observed (Appendix A). A similar trend was observed for absorbance bands, with Figure 3D representatively showing the absorbance spectra and maxima of **1** overlapping at pH 1 and 4 at 694 nm, whereas, at pH 7, the absorbance maximum is 751 nm (Table 1). Fluorophores **2** and **3** showed similar absorbance profiles (Appendix A). Notably, these results revealed that the differing conjugated groups of **1**–**3** did not significantly change the spectroscopic properties. Encouragingly, the mean 68 (±4) nm wavelength separation of the phenolate/phenol emissions for **1**–**3** would be more than sufficient to distinguish one from the other in a microscopy setting.

The responsive interplay between the two emissions was illustrated by pH titrations of **1**–**3**. At neutral pH, only the longest wavelength emission was noted, though this steadily decreased as the pH was lowered and the shorter wavelength band emerged. Upon titration between pH 7 and 2, it was seen that the interchange of both emissions occurred across an important intracellular acidity range of pH 5 to 4 with apparent pKa values of **1**–**3** measured as 4.5, 4.3 and 4.1, respectively (Figure 4, panels A–C). These values were particularly encouraging in the context of live cell imaging as the neutral plasma membrane would be predicted to be emissive only in the 810 (±40) nm region, as would early endosomes (pH~6.3) and other non-acidic vesicles. At the lower pH window of 4 to 5, the crossover points of green and red titration profiles occur, indicating that both emissions in the 810 (±40) and 715 (±25) nm wavelength ranges would be observable. In a cellular context, this range is comparable to that found in lysosomes and LAVs. Areas of red-only emission would not be expected, as subcellular regions with pH below 4 are not found in mammalian cells. The conjugates showed relatively minor changes with solvent polarities [28] and viscosities, indicating the pH response would be dominant (Appendix A). Taken together, all the photophysical properties were notably consistent for both peptide and PEG conjugates, showing the potential for broader fluorochrome imaging use.

### 2.2. Live Cell Microscopy

Human epithelial breast cancer MDA-MB 231 cells were chosen to study the potential for simultaneously utilizing the dual emissions. As a triple-negative cell line, they are considered to be aggressively invasive and are known to express membrane integrins, including αvβ3 and αvβ5 [41,42,43,44]. The metastatic nature of these cells is mediated by proteolytic degradation of the extracellular matrix (ECM). MDA-MB 231-derived tumors are known to effectively acidify the surrounding ECM and have LAVs in which endocytosed extracellular matrix can be digested by activated lysosomal proteinases such as cathepsin [45].

Taking guidance from the photophysical results described above, two spectral emission ranges of 690–740 nm and 770–850 nm were selected to use as the acquisition channels on the spectral confocal microscope, to which the pseudo colors of red and green were assigned, respectively (Figure 4 panel D). In order to test for simultaneous observation of emissions from both the phenol and phenolate species of the fluorophores in vitro, live cells were incubated with 2 µM of **3** with test images acquired at 5 min and 5 h, using the two different wavelength detection ranges to observe both on-states. In Figure 5 panel A, it can be noted that at 5 min, the plasma membrane is distinctly stained, being the first part of the cell to interact with the fluorophore upon incubation. The membrane, being at or close to neutral pH, is stained green, which is the color attributed to the phenolate species that should predominate, being sensed by the confocal microscope detector set between 770 and 850 nm (Figure 5, panels A,C). By collecting an emission spectrum of the plasma membrane at this time point, it was confirmed that the emission maximum was 790 nm, similar to what was observed in a pH 7 DMEM solution (787 nm) (Figure 5, panel G). At 5 min, it is important to note that no emission was detectable in the red channel (690–740 nm), likely because the fluorophore had not yet time to internalize and concentrate inside the cell (panel B). However, at 5 h in the green channel, both the plasma membrane and internal vesicles were imageable (panel D). In the red channel for the same field of view, vesicles were also observed with acidic lumens due to the existence of the phenol on-state (panel E). Upon collecting an emission spectrum (from 650 to 850 nm) of several of these vesicles, an emission max at 720 nm was obtained consistent with the phenol state emission profile observed in the cuvette experiments at pH 4 (720 nm) (Figure 5, panel H). Yet, within this profile, a second longer wavelength emission band was also recorded with λ-max of 787 nm, showing that both on-states are observable in these subcellular locales (panel H). The red and green overlaid image in panel F confirmed that green and red emissions were co-compartmentalized within the same vesicles. This observation of a two-color emission from the acidic vesicles is consistent with a lysosomal pH between 4.2 and 5.0, which is close to the intersecting point of both emissions in the pH titration curves of Figure 4.

To further develop an understanding of the intracellular progression of all three conjugates **1**–**3**, each was incubated with live cells under identical conditions, with images acquired using the same parameters at three different time points of 30 min, 2 h, and 5 h. At 30 min for each fluorophore, the cell membrane was intensely stained green, with only a few vesicles observed (Figure 6, panel A). At 2 and 5 h, the number of red and green co-stained acidic vesicles had significantly increased, with a significant number observable in both spectral windows (panels B and C). As the overall progression from the plasma membrane to vesicle internalization was similar for all three fluorophores, showing little difference for the peptide or pegylated derivatives, variances in uptake pathways could not be distinguished. For other representative images, see Appendix A and Z-stack Appendix A.

To exclude the possibility that the overlap between the two channels within some vesicles was due to the reservoir of the fluorophore in the media, which is continually being internalized over time, this image sequence was repeated using a pulse-chase experimental approach. In this way, cells were treated with 2 µM of fluorophore at 37 °C for 30 min, following which the media was removed, and cells were washed with pre-warmed PBS to remove any unbound fluorophore and fresh media added. Images taken at different time points over a 5 h period also showed dual emissive vesicles (Appendix A).

The final stage of the fluorophore cellular journey would be its efflux from cells back into the media. As the outflow of material from cells can occur both through exocytosis and within released EVs, it was of interest to determine whether it was contained within the EVs emanating from cells and what emission characteristics would be observed. In previous work, we have shown that a bis-alkylsuphonic acid substituted by BF_2_-azadipyrromethane was contained within EVs upon its release by cells [46]. As such, following a 2 h incubation of cells with **1**, the cell media was replaced, and incubation continued for 48 h, allowing sufficient time for intracellular trafficking and efflux to occur. Cell-released EVs were isolated and purified from cellular debris through centrifugation and size exclusion filtration procedures. Nanoparticle tracking analysis (NTA) showed a distribution of EV sizes from 100 to 400 nm, with confocal microscopy establishing that isolated EVs were emissive (Figure 7, panel A). Upon further spectroscopic and microscopic analysis, EVs were confirmed as green fluorescent stained with an emission spectrum correlating with that of the neutral pH phenolate form with λ_max_ at 788 nm (Figure 7, panel B). Upon microscopy imaging of the cells following the 48 h efflux incubation time, it was of interest to note that some fluorophore remained in the cells in the LAVs, which upon spectral analysis were both red and green emissive (panel C, for other representative images, see Appendix A).

### 2.3. Image Analysis

Statistical evidence of co-compartmentalization of the phenolate and the phenol states of **1**–**3** was obtained through calculated Pearson’s and Manders’ coefficients of images taken at 2 and 5 h. Using the pH titration curves and pKa values as a guide, it could be anticipated that early trafficking vesicles of pH above 6 would be phenolate green only, whereas more acidic vesicles of pH 4–5 would be both red and green (Figure 4 panel D). Though it should be acknowledged that some differences may exist between the cuvette solution measurements and vesicles, the trends would be expected to be comparable. Good Pearson’s coefficient values ranging from 0.75 (±0.05) to 0.52 (±0.09) were obtained at both time points, indicative of considerable but not complete co-localization (Table 2). This indicated that many but not all vesicles were dual emissive. It was notable that Pearson’s values for **1**–**3** at both time points were close to those obtained for the Manders’ M1, which (as it is for green emission over red) implies that some vesicles were only green emissive. This was supported by the M2 (red over green) values, which were all higher than those of M1, ranging from 0.98 (±0.01) to 0.78 (±0.05), indicating that vesicles that were red emissive were also green emissive (Table 2).

Using a software image color subtraction function, it was possible to distinguish and visualize non-acidic vesicles within cells, as shown in the image sequence of Figure 8 [47,48]. Representatively, cell images were taken following a 2 h incubation with **1**, as previously described in both green and red channels. After creating and applying a mask to both channels in Image J, areas of red pixels were subtracted from the green image (panel A minus panel B), resulting in an image of areas that are only green fluorescent. This showed the areas within the cell cytoplasm that were green only (in addition to the plasma membrane), which would be vesicles of pH above 6. Additional examples are shown in the Appendix A.

### 2.4. Fluorescence Lifetime Imaging Microscopy

An added advantage of the dual emissive on-to-on fluorophore design is that the dynamic interrelationship between both states can also be scrutinized through their different lifetimes using FLIM and phasor plot data analysis [49]. Phasor plot analysis allows for the graphical representation of the raw fluorescence lifetime data, such that each pixel in a FLIM image is transformed to a point in the phasor plot, with pixels containing a combination of two different lifetimes graphed according to the weighted linear combination of their contributions [50]. This makes phasor plot analysis ideally suited for tracking pH-responsive molecules such as **1**–**3** that can dynamically interconvert between both phenolate and phenol states in relation to the pH of their microenvironment.

In order to characterize the lifetimes of the phenol and phenolate states of the fluorophore, DMEM solutions of **3** at pH 2, pH 4, and pH 7 were prepared, and the fluorescence phasor lifetimes were obtained. The values measured from lowest to highest pH were 1.2, 0.9, and 0.7 ns, respectively (Appendix A). This predicts that the shorter lifetime of ~0.7 ns emanating from the phenolate state would be observable in the neutral plasma membrane, with longer average lifetimes arising from more acidic vesicles. Experimentally, cells were incubated with **1** and FLIM images taken at 30 min, 2 h, and 5 h, matching the same time points used for confocal images (Figure 9, panels A and B at each time point). As anticipated, at 30 min, the neutral plasma membrane had a shorter phasor lifetime and was the most intense photon distribution region of the phasor plot consistent with it being first stained (30 min; panel C). At 2 h and 5 h, the photon intensity map could be seen to shift to longer lifetimes as more fluorophore is internalized within vesicles of lower pH (2 h and 5 h; panel C). Phasor separation of the FLIM images based on shorter (0.7 ns, green circle) and longer (1.2 ns, red circle) lifetimes gave two individual images showing the subcellular regions matching these lifetimes (Figure 9, panels D, E). These images and the shift in the intensity of the phasor plot photon distribution towards longer lifetimes are particularly distinct at 5 h, illustrating how the transport trajectory of the fluorophore from membrane to vesicles can be tracked over time (30 min, 2 h, 5 h; panel C). For representative FLIM time course imaging with **1** and **2**, see Appendix A.

## 3. Experimental

### 3.1. General

Fluorophores **1** and **3** were synthesized following the literature procedures [29]. Phosphate buffered saline (PBS), Dulbecco’s modified Eagle’s medium (DMEM), 1% l-glutamine, fetal bovine serum (FBS), and 1% penicillin–streptomycin were purchased from Thermo Fisher Scientific, Dun Laoghaire, Dublin, Ireland. *c*RAD peptide was purchased from Peptides International Louisville, Kentucky, USA. MDA-MB 231 cells were purchased from ATCC. HPLC grade water was used and purchased from Sigma-Aldrich Arklow, Wicklow, Ireland and filtered using an MF-millipore from 33 mm filter Sigma-Aldrich Arklow, Wicklow, Ireland,. Absorbance spectra were recorded with a Varian Cary 50 scan ultraviolet–visible spectrometer. Emission/excitation spectra were recorded on a FluoroMax Plus spectrofluorometer. Confocal and FLIM images were acquired using the Leica Stellaris 8 Falcon system (objective: Leica HC PL APO CS2 100X/1.40 oil immersion) Wetzlar, Germany. The white light laser was used to excite the fluorophores. Images were processed using LASX Falcon (FLIM) software (version 4.6.0.27096) and ImageJ v1.54f.

### 3.2. Synthesis and Characterization

Synthesis of 2-((2S,5R,8S,11S,14S)-5-benzyl-8-(1-(4-(5,5-difluoro-7-(4-hydroxy-3-nitrophenyl)-1,9diphenyl-5H-5λ^4^,6λ^4^-dipyrrolo[1,2-c:2′,1′-f][1,3,5,2]triazaborinin-3-yl)phenoxy)-2,11,20-trioxo-6,9,15,18-tetraoxa-3,12,21-triazapentacosan-25-yl)-11-(3-guanidinopropyl)-14-methyl-3,6,9,12,15-pentaoxo-1,4,7,10,13 pentaazacyclopentadecan-2-yl)acetic acid **2**. Following a similar previously reported procedure for the synthesis of **1** [28] (Appendix A) a vial containing cyclic peptide cyclo[Arg-Ala-Asp-D-Phe-Lys(PEG-PEG)] (5 mg, 0.0056 mmol) was flushed with N_2_ and its contents were dissolved with 600 μL anhydrous DMSO. The vial was briefly sonicated, and its contents were added via syringe to an oven-dried N_2_ flushed 1.5 mL vial containing the activated ester fluorochrome precursor (4.5 mg, 0.0062 mmol) [41]. The vial originally containing the peptide was rinsed with an additional 100 μL of anhydrous DMSO, with this also added to the reaction vial, and the reaction stirred at rt with its progression followed by HPLC (ACN:H_2_O 60:40 with 10 mM NH_4_HCO_3_—flow 0.6 mL/min). After 4 h, the reaction was judged to have gone to completion, and the reaction mixture was diluted with H_2_O (10 mL) and lyophilized. The crude was dissolved in 40:60; ACN:H_2_O with 10 mM NH_4_HCO_3_, filtered through a PTFE 0.45 μM syringe filter, and the resulting green solution was purified by reverse phase semi-prep chromatography (YMC Triart Phenyl, 10 × 150 mmI.D. S-5 μm, injection volumes 600 μL—eluant ACN:H_2_O 40:60—flow 3 mL/min). Pure fractions (analyzed by HPLC) of **2** were combined and reduced to dryness by lyophilization to afford **2** as a light green powder (7.8 mg, 90%). ^1^H NMR (500 MHz, DMSO-*d*_6_) δ: 12.17 (s, 1H), 8.76 (s, 1H), 8.27 (dd, *J* = 9.0, 2.2 Hz, 1H), 8.24–8.09 (m, 10H), 8.00 (d, *J* = 7.1 Hz, 1H), 7.90 (dd, *J* = 47.0, 8.6 Hz, 1H), 7.68 (m, 3H), 7.62 (s, 1H), 7.54 (t, *J* = 7.4 Hz, 5H), 7.48 (t, *J* = 7.3 Hz, 2H), 7.23 (t, *J* = 7.3 Hz, 2H), 7.15 (dd, *J* = 17.0, 7.7 Hz, 6H), 4.66 (s, 2H), 4.42 (ddd, *J* = 25.0, 14.3, 7.6 Hz, 2H), 4.12–4.02 (m, 2H), 3.95–3.90 (m, *J* = 15.5 Hz, 1H), 3.89 (s, 2H), 3.85 (s, 2H), 3.60–3.53 (m, 8H), 3.50 (t, *J* = 5.7 Hz, 2H), 3.45 (t, *J* = 6.0 Hz, 2H), 3.30–3.25 (m, 3H), 3.09 (d, *J* = 7.0 Hz, 2H), 3.02 (dd, *J* = 13.3, 6.9 Hz, 2H), 2.94 (dd, *J* = 13.3, 7.9 Hz, 1H), 2.79 (m, 2H), 2.44 (dd, *J* = 16.1, 6.0 Hz, 1H), 1.62 (dd, *J* = 38.1, 28.8 Hz, 3H), 1.39 (m, 5H), 1.24 (d, *J* = 7.1 Hz, 3H), 1.04 (d, *J* = 6.6 Hz, 2H) ppm (Appendix A). HRMS calcd. for C_74_H_87_BN_15_O_18_F_2_ [M+H]^+^: 1522.6415; found 1522.6458 Da.

### 3.3. Spectroscopic pH Analysis

Individually, compounds **1**, **2**, and **3** were accurately weighed and dissolved in PBS of appropriate volume to yield stock solutions (250 μM) of each. Stock solutions were diluted to a concentration of 5 μM with complete Dulbecco’s modified Eagle’s medium (DMEM), containing 10% FBS and triton X-100 (0.34 mM). The pH of the mixed solutions was adjusted, and diluted HCl (0.1 M) or NaHCO_3_ (1 M) was added to create solutions ranging from pH 2 to 8. Absorbance and fluorescence spectra were recorded (excitation 625 nm, slits excitation 5 nm, emission 5 nm; excitation 700 nm, slits excitation 5 nm, emission 5 nm). Scans were carried out on all aqueous fluorophore solutions at room temperature.

Repeatability of pH responses:

**1**, **2**, and **3** were each dissolved in PBS at a concentration of 5 µM. Solution pH was cycled between 2 and 8 using aqueous HCl and KOH, with emissions recorded each time.

Polarity and viscosity fluorescence influences:

Fluorophores **1**, **2**, and **3** were each dissolved in toluene, tetrahydrofuran, dimethylformamide, and DMSO at a concentration of 5 µM, containing trifluoroacetic acid, and their emission spectra were recorded. Solutions (5 µM) of **1**, **2**, and **3** were prepared in five different ratio mixtures of ethylene glycol and glycerol to give the required viscosities between 25 and 400 mPa s, and their emission spectra were recorded.

### 3.4. Cell Culture and Live Cell Confocal and FLIM Microscopy

Triple-negative breast cancer MDA-MB-231 cells were cultured in DMEM supplemented with 1% l-glutamine, 1% penicillin–streptomycin, and 10% FBS at 5.0% CO_2_ and 37 °C. Cells were seeded on an eight-well Ibidi chamber slide at a density of 1 × 10^4^ cells per well and allowed to proliferate for 48 h before imaging. Confocal images were acquired using the Leica Stellaris 8 Falcon system (objective: Leica HC PL APO CS2 100X/1.40 oil immersion). The white light laser was used to excite the fluorophores at 10% of the power. Images were processed using LASX Falcon (FLIM) software (version 4.6.0.27096) and ImageJ. ImageJ plugin JACoP was used with the default settings at all parameters to analyze the degree of colocalization (Manders’ overlap coefficients and Pearson’s correlation coefficient). Illumination conditions for each fluorophore were as follows: **1**, **2**, and **3** excited at 680 nm (emission collected using a HyD X detector from 690 to 740 nm) and also at 760 nm (emission collected using a HyD R detector from 770 to 850 nm). Using FLIM, the stop condition for photon accumulation was set to 2000 photons in the brightest pixel; the white light laser was set at 680 nm to excite **1**, **2**, and **3**, and minimum power was used to avoid the detector saturation. The HyD R detector was used to collect the emission between 690 and 850 nm. A scanning speed of 200 Hz was used for all images. Emission scans produced by white light laser (10%) were also taken. Experiments were repeated in triplicate, and average values were used.

Slides were placed on the microscope stage, and a field of view consisting of a suitable selection of cells was chosen. Cell media from the well was aspirated and replaced with 300 µL of cell media containing **1**, **2**, or **3** (2 µM), and the imaging protocol was started. For the pulse-chase experiments, cells were incubated with **1**, **2**, or **3** (2 µM) at 37 °C and 5.0% CO_2_ for 30 min. These cells were then washed three times with pre-warmed PBS to remove any unbound fluorophore, and fresh, complete media was added to the wells prior to imaging. An incubator system controlled environmental temperature and CO_2_% (5%) during all live experiments.

### 3.5. EV Isolation

The conditioned medium was pre-clarified by passing the sample through a 0.45 µm sterile filter to remove cells and debris. The filtered sample was concentrated by using an Amicon^®^ Ultra (AU) filter (100 kDa MWCO). Extracellular vesicles were enriched from serum-free cell culture media as follows: 2 mL of sterile PBS was added to the AU filter and centrifuged at 4000× *g* for 10 min in a swinging bucket rotor. PBS was removed from the bottom of the filter device, and the filtrate was aspirated from the collection tube. A total of 15 mL of pre-clarified sample was added to the AU filer and centrifuged at 4000× *g* for 30 min. The collection tube was emptied, and the contents of the filter device were supplemented with 14 mL of pre-warmed PBS and gently pipetted several times. This was then centrifuged at 4000× *g* for 30 min. The concentrated sample recovered from the filter device contains washed extracellular vesicles.

### 3.6. ImageJ Protocol for Color Subtraction

The red channel image and green channel images were opened in ImageJ. Select the red channel image. Go to Image > Adjust > Threshold. Accept the default threshold settings and apply. On the same red channel image, go to Edit > Invert. Once inverted, go to Process > Math > Divide, enter “255” in Value, and click “OK”. Go to Process > Image Calculator and multiply the resulting red channel image by the green channel image. Select “Create New Window”.

## 4. Conclusions

The pathway of peptide and polyethylene glycol conjugated bio-responsive fluorophores from the first encounter with the plasma membrane through to release within EVs has been achieved by exploiting their unique dual NIR emission and lifetime characteristics. Irrespective of the conjugate groups bonded to the fluorochrome, the spectral properties from solution cuvette measurements across the different fluorophores remained similar to each other, and this was reflected in live cell imaging experiments. In the neutral pH environments of the plasma membrane, early trafficking vesicles, and EVs, a single emission with a λ_max_ of ~790 nm was recorded. In contrast, within intracellular vesicles of lower pH, such as lysosomes and LAVs, a double emission with maxima at ~720 and ~790 was notable. This demarked these vesicles as having pHs in the 4–5 range corresponding to the pKa values of the fluorophores. Similarly, phasor plot analysis of FLIM experiments established that in addition to the fluorophore emitting at two spectrally different wavelengths, the two forms of the molecule also possess different phasor lifetimes (0.7 ns for the phenolate and 1.2 ns for the phenol). The shift over time from shorter to longer lifetimes could be used to time-track fluorophore progression through cells. In light of the demonstrated capabilities of this fluorochrome, its dual emissive nature and informative lifetime capacity can be used in tandem to interrogate molecular functions, interactions, and microenvironments of cells according to changes in pH.

## Figures and Tables

**Figure 1 molecules-29-02474-f001:**
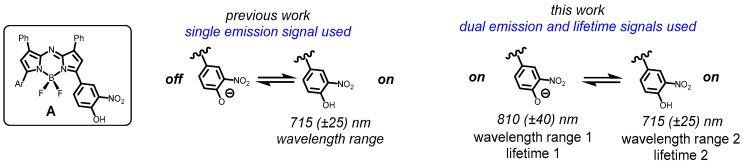
General structure of pH-responsive fluorophore A. Prior work using a single emission off-to-on format and current goals of exploiting both distinct interchangeable emissions in a dual emission on-to-on manner for intensity and lifetime imaging.

**Figure 2 molecules-29-02474-f002:**
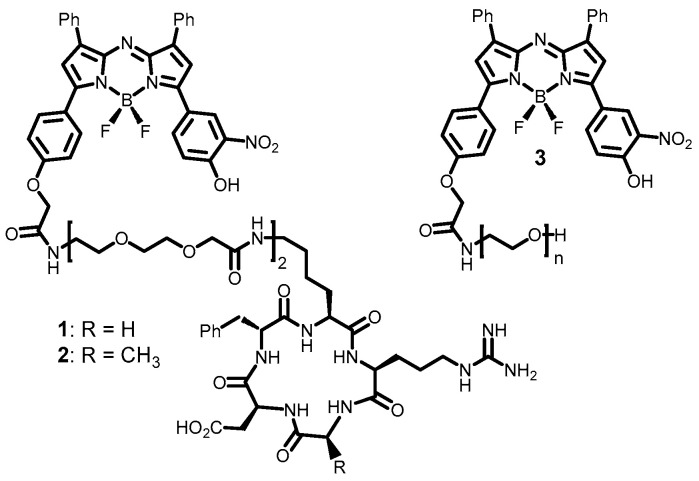
Structures of double emissive bio-responsive fluorophores **1**–**3** used in this study.

**Figure 3 molecules-29-02474-f003:**
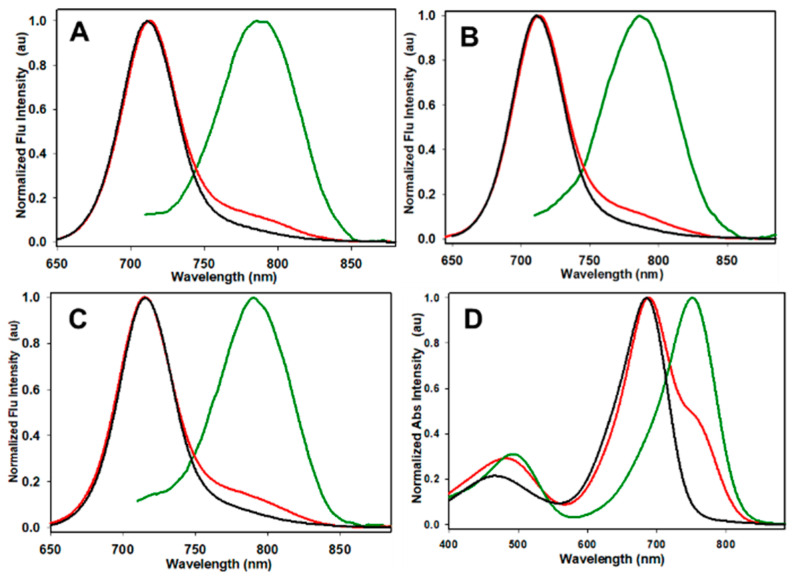
Spectroscopic characteristics of **1**–**3** at pH 7 (green), pH 4 (red), and pH 1 (black) in DMEM at 5 µM concentration. (**A**) Emission spectra of **1**. (**B**) Emission spectra of **2**. (**C**) Emission spectra of **3**. (**D**) Absorbance spectra of **3**.

**Figure 4 molecules-29-02474-f004:**
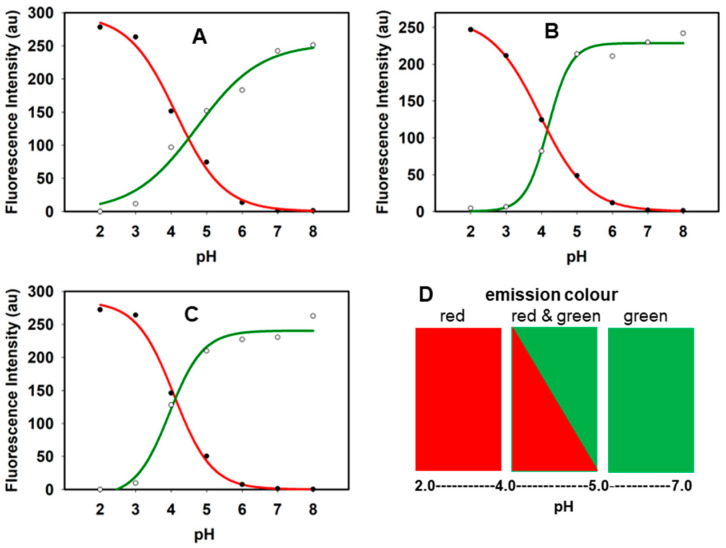
Fluorescence—pH titration responses for **1**, **2**, and **3** in DMEM. (**A**) Changes in emissions for **1** showing decreasing band at λ_max_ 790 nm (green, open circles) and increasing band intensity at λ_max_ 720 nm (red, filled circles) from pH 8 to 2. (**B**) Changes in emissions for **2** showing decreasing band at λ_max_ 790 nm (green) and increasing band intensity at λ_max_ 720 nm (red) from pH 8 to 2. (**C**) Changes in emissions for **3** showing decreasing band at λ_max_ 790 nm (green) and increasing band intensity at λ_max_ 720 nm (red) from pH 8 to 2. (**D**) Correlation of spectroscopy titration data with microscopy emission acquisition parameters; above pH 5 single emission band of the wavelength range of 770—850 nm (pseudo color green); between pH 4 and 5, both emission bands of wavelength ranges of 690–740 nm (pseudo color red) and 770–850 nm; below pH 4 single emission band of wavelength range 690–740 nm.

**Figure 5 molecules-29-02474-f005:**
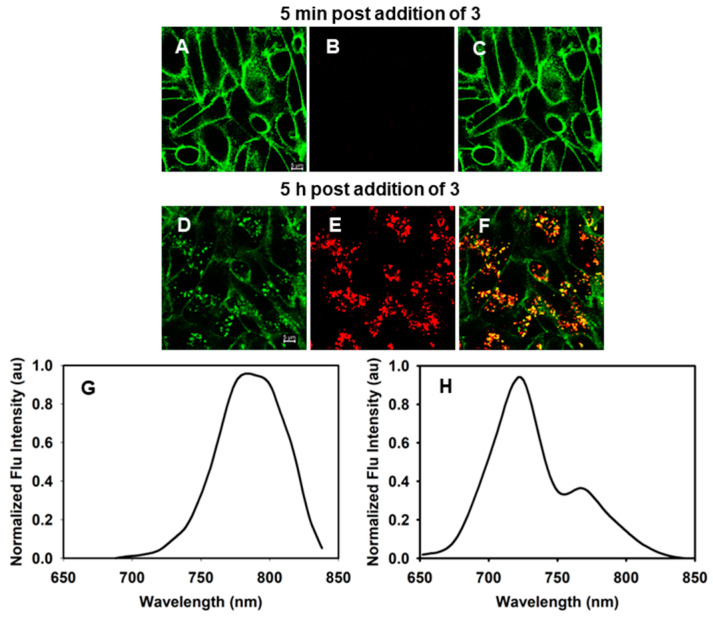
Confirmation of dual fluorescence detection from in live MDA-MB-231 cells following their treatment with 2 µM of **3** and imaged at 5 min and 5 h. (**A**) Cells imaged using 760 nm excitation and 770–850 nm detector collection 5 min post-treatment. (**B**) Cells imaged using 680 nm excitation and 690–740 nm detector collection 5 min post-treatment. (**C**) Overlay of images (**A**,**B**). (**D**) Cells imaged using 760 nm excitation and 770–850 nm detector collection 5 h post-treatment. (**E**) Cells imaged using 680 nm excitation and 690–740 nm detector collection 5 h post-treatment. (**F**) Overlay of images (**D**,**E**). (**G**) Emission spectrum of cell plasma membrane region from image (**A**). (**H**) Emission spectrum of cell vesicle region from image (**E**). Scale bar: 5 μM.

**Figure 6 molecules-29-02474-f006:**
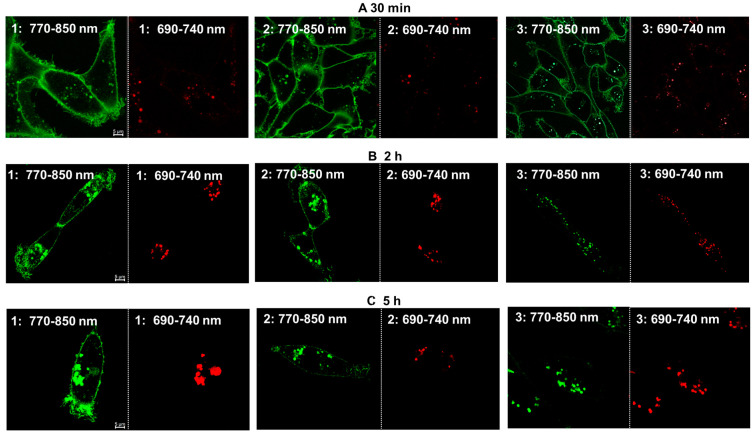
Time course of dual color imaging of live MDA MB 231 cells following treatment with 2 µM concentration of **1** (left), **2** (middle), and **3** (right) at (**A**) 30 min, (**B**) 2 h, and (**C**) 5 h. Images acquired using excitation at 760 nm with a collection from 770–850 nm (green) or excitation at 680 nm and collection between 690–740 nm. Scale bar: 5 μM.

**Figure 7 molecules-29-02474-f007:**
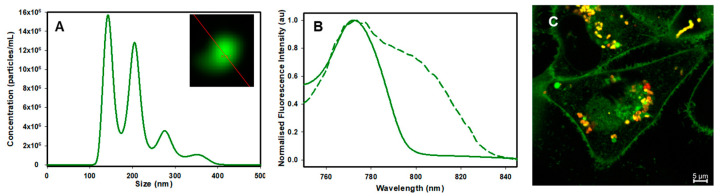
Spectroscopic and microscopic analysis of isolated EVs from MDA-MB 231 cells treated with **1** and remaining cells after efflux. (**A**) NTA analysis of isolated EVs with inset showing microscopy image of a single EV. (**B**) Emission spectra of isolated EVs taken on a fluorimeter (solid trace 0.1 cm cuvette pathlength, excitation 740 nm, slit widths 5 nm) and microscope (dashed line). (**C**) Confocal image of cells following 48 h efflux. Scale bar: 5 μM.

**Figure 8 molecules-29-02474-f008:**
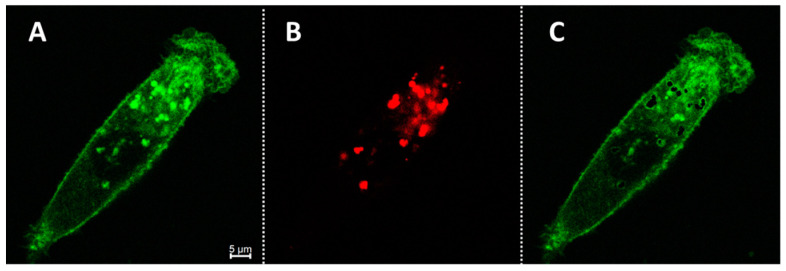
Visualization of intracellular non-acidic vesicles via image subtraction in live MDA MB 231 cells following treatment with 2 µM **1** for 2 h. (**A**) Image acquired using excitation at 760 nm with a collection from 770 to 850 nm (green). (**B**) Image acquired using excitation at 680 nm and collection between 690 and 740 nm (red). (**C**) Resulting in imaging following the subtraction of image (**B**) from (**A**). Scale bar: 5 μM.

**Figure 9 molecules-29-02474-f009:**
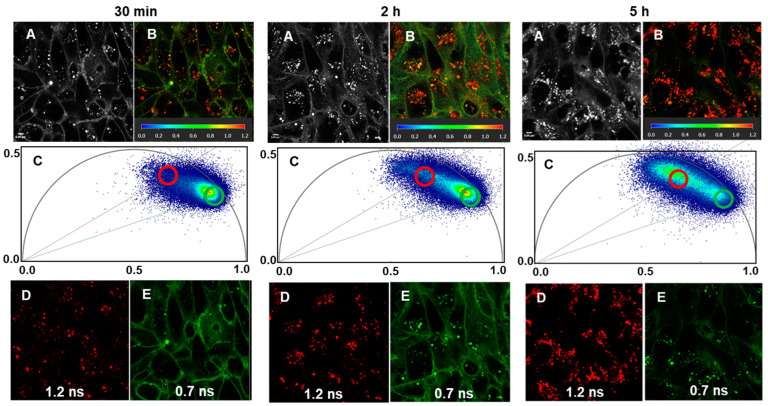
FLIM imaging with phasor analysis of live MDA MB 231 cells following treatment with 2 µM of **3** at 30 min, 2 h, and 5 h. (**A**) Intensity images at specific time points (**B**) FLIM image with lifetime heat map at specific time points. (**C**) Phasor plots at specific time points with phasor lifetimes of 1.2 ns (red circle) and 0.7 ns (green circle) are shown (**D**,**E**). Scale bar: 5 μM.

**Table 1 molecules-29-02474-t001:** Spectroscopic data for **1**–**3** in complete Dulbecco’s modified Eagle’s medium.

Fluorophore Conjugate	λ_max_ abs/nm pH 7 (pH 4)	λ_max_ flu/nm pH 7 ^a^ (pH 4 ^b^)	pKa ^c^
**1**	751 (694)	792 (719)	4.5
**2**	750 (688)	783 (719)	4.3
**3**	754 (692)	787 (720)	4.1

^a^ Excitation at 700 nm; ^b^ excitation at 625 nm; ^c^ determined by fluorescence titration.

**Table 2 molecules-29-02474-t002:** Pearson’s coefficients and Manders’ coefficients (M1, M2) for **1**–**3**
^a^.

Fluorophore Conjugate	Time (h)	Pearson’s Coefficient	M1	M2
**1**	2	0.7 (±0.06)	0.78 (±0.03)	0.96 (±0.02)
**1**	5	0.75 (±0.05)	0.76 (±0.09)	0.98 (±0.01)
**2**	2	0.6 (±0.02)	0.59 (±0.04)	0.96 (±0.01)
**2**	5	0.65 (±0.09)	0.68 (±0.1)	0.93 (±0.03)
**3**	2	0.52 (±0.09)	0.53 (±0.11)	0.78 (±0.05)
**3**	5	0.69 (±0.03)	0.73 (0.08)	0.90 (±0.02)

^a^ Mean of triplicate measurements of three independent experiments ± the standard deviation. M1: green over red. M2: red over green.

## Data Availability

Data are contained within the article and Appendix A.

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
