# Peer review of "A NIR-Fluorochrome for Live Cell Dual Emission and Lifetime Tracking from the First Plasma Membrane Interaction to Subcellular and Extracellular Locales"

_molecules, 2024, doi:10.3390/molecules29112474_

Round 1
Reviewer 1 Report
Comments and Suggestions for Authors
The authors reported synthesis of dual-emissive probe and its application on bio-imaging. The pH tuned dual-emission were successfully used for exploring cellular molecular functions, interactions, and microenvironments and provides new tools for cell biology research. The science is clear with conclusion well supported by the data. The manuscript was well-organized. Therefore, it was recommended for publishing after minor revision.
1. The introduction of phenol groups significantly changed fluorescence behaviors of BODIPY. The fluorescent properties of the probe should be given, like its quantum efficiency, extinction coefficient and photostability, as well as its phototoxicity, etc.
2. The fluorophores were functionalized with PEO and peptide. However, the specific pathway, especially the roles of PEO and peptide, by which the probes traverse the cell membrane to reach secondary cellular organelles is not clearly elucidated. Moreover, the similar probes have been already reported in their previous work (ref.41), it is necessary to address the importance and significance of novel probe 2 comparing with the previous Probe 1 and 3.
Author Response
Reviewer 1 Comments:
Thanks for the supportive comments and constructive feedback on our manuscript.
Response to comment #1:
The quantum efficiency, extinction coefficient and photostability, and phototoxicity were not included as they have been previously reported in the cited reference #28. This has been made clearer in the manuscript by including additional text stating; “Previously, we reported a BF2azadipyrromethene of general structure A as a lysosomal acidity off-to-on responsive probe with
excellent photostabilities, good quantum yields, and no phototoxicity at concentrations used for imaging (Figure 1).[28]”
Response to comment #2:
The specific cell entry pathways were not the focus of this work but to clarify further additional text has been include in the “live cell microscopy” section of he manuscript stating that “As the overall progression from plasma membrane to vesicle internalisation were similar for all three fluorophores showing little difference for the peptide or pegylated derivatives, variances in uptake pathways could not be distinguished.” Probe 2 was included in the study to confirm that
no discernible difference were observed between peptides conjugates and PEG conjugates in imaging experiments.
Reviewer 2 Report
Comments and Suggestions for Authors
Please check the below comments,
1. There is a rotatable single bond in the fluorophore. The viscosity of the cell can be adjusted by physiological or pathological processes can change. Experiments should be designed to verify that the viscosity has no effect on the fluorescence intensity of the molecule.
2. Experiments should be designed to prove that solvent polarity has no effect on molecular fluorescence intensity.
3. The photophysical properties of the fluorescent probe change based on the reversible change of the phenolic hydroxyl proton. After multiple cycles, whether it can maintain the same fluorescence intensity and fluorescence lifetime.
4. The nuclear magnetic resonance hydrogen spectrum, nuclear magnetic resonance carbon spectrum and mass spectrum of molecule A and fluorophore 1-3 were not shown in the text and support information.
5. The article lacks a schematic diagram of the synthetic route
Comments on the Quality of English LanguageExtensive editing of English language required.
Author Response
Reviewer 2 Comments:
Thanks for the supportive comments and constructive feedback on our manuscript.
Response to comment #1:
New experimental data showing the minor effect on fluorescence for different viscosities has been added to the supporting information (Figure S3) and discussed in the manuscript with the addition of new text stating “The conjugates showed relatively minor changes with solvent polarities[28] and viscosities indicating the pH response would be dominant (SI Figure S3).”
Response to comment #2:
New experimental data showing the minor effect on fluorescence intensity for different solvent polarities has been added to the supporting information (Figure S3) and discussed in the manuscript with the addition of new text stating “The conjugates showed relatively minor changes with solvent polarities[28] and viscosities indicating the pH response would be dominant (SI Figure S3).” Similar data had been previously published in reference # 28 which has been cited.
Response to comment #3:
New experimental data showing multiple cycles of the reversible phenol / phenolate forms of the fluorophores has been included in the supporting information (Figure S1) and discussed in the manuscript with the addition of new text stating “When the fluorophores were subjected to repeat cycles of acidification and neutralization, no loss in fluorescence intensity was observed (SI Figure S1)”.
Response to comment # 4:
The NMR and mass spectra for fluorophores 1 and 3 have been previously published in reference # 29 which has been stated in the experimental section as follows: “Fluorophores 1 and 3 were synthesized following literature procedures [29]”. The synthetic scheme for the synthesis of fluorophore 2 has been added to the supporting information as Figure S10. The NMR, mass spectra, and HPLC trace for fluorophore 2 have now been added to the supporting information in Figure S11.
Response to comment #5:
As the synthetic route followed the previously published route, a schematic diagram has been added to the supporting information as Figure S11.
Additional comment:
The English language has been carefully checked and improved throughout the manuscript.